# Revealing hole trapping in zinc oxide nanoparticles by time-resolved X-ray spectroscopy

Thomas J. Penfold[1], Jakub Szlachetko[2,3], Fabio G. Santomauro[4], Alexander Britz[5,6], Wojciech Gawelda [5,7], Gilles Doumy[8], Anne Marie March[8], Stephen H. Southworth[8], Jochen Rittmann[4], Rafael Abela[2], Majed Chergui [4] & Christopher J. Milne [2]

Nanostructures of transition metal oxides, such as zinc oxide, have attracted considerable interest for solar-energy conversion and photocatalysis. Both applications are sensitive to the transport and trapping of photoexcited charge carriers. The probing of electron trapping has recently become possible using time-resolved element-sensitive methods, such as X-ray spectroscopy. However, valence-band-trapped holes have so far escaped observation. Herein we use X-ray absorption spectroscopy combined with a dispersive X-ray emission spectrometer to probe the charge carrier relaxation and trapping processes in zinc oxide nanoparticles after above band-gap photoexcitation. Our results, supported by simulations, demonstrate that within 80 ps, photoexcited holes are trapped at singly charged oxygen vacancies, which causes an outward displacement by ~15% of the four surrounding zinc atoms away from the doubly charged vacancy. This identification of the hole traps provides insight for future developments of transition metal oxide-based nanodevices.

[1] Chemistry—School of Natural and Environmental Sciences, Newcastle University, Newcastle upon Tyne, NE1 7RU, UK. [2] SwissFEL, Paul Scherrer Institut, CH-5232 Villigen-PSI, Switzerland. [3] Institute of Nuclear Physics, Polish Academy of Sciences, 31-342 Kraków, Poland. [4] Ecole polytechnique Fédérale de Lausanne, Laboratoire de spectroscopie ultrarapide, ISIC, FSB and Lausanne Centre for Ultrafast Science (LACUS), CH-1015 Lausanne, Switzerland. [5] European XFEL, Holzkoppel 4, D-22869 Schenefeld, Germany. [6] The Hamburg Centre for Ultrafast Imaging, Luruper Chaussee 149, 22761 Hamburg, Germany. [7] Faculty of Physics, Adam Mickiewicz University, Umultowska 85, 61-614 Poznań, Poland. [8] Argonne National Laboratory, 9700 S. Cass Ave., Argonne, IL 60439, USA. Correspondence and requests for materials should be addressed to C.J.M. (email: chris.milne@psi.ch)

D evices for solar-energy conversion[1] and photocatalysis[2] based on transition metal oxide (TMO) nanoparticles (NPs) and mesoporous films are amongst the most promising routes for addressing our escalating energy demands. These rely on sunlight to generate charge carriers and consequently, it is crucial to understand the room temperature (RT) creation, transport, and trapping of the photogenerated holes and electrons over a wide distribution of time and length scales. Yet, the dynamics of these processes remain poorly understood, in part due to the complex interplay of structural and electronic effects that affect the various relaxation pathways.

The most promising TMO solar devices use nanostructures of titanium dioxide[3] ($TiO_2$) and zinc oxide[4] (ZnO). While historically $TiO_2$ has been the most prominent, ZnO has also attracted significant interest owing to an electron mobility two orders of magnitude larger than $TiO_2$[5]. In addition, ZnO can be easily synthesised in a wide variety of nanostructures[4], which enables control of its properties via the NPs' size and shape. However, two limitations have restricted the application of ZnO in solar energy conversion. First, rapid recombination of photo-generated charge carriers[6] results in short-lived electrons and holes (~1–2 ns), limiting device efficiency. Second is its instability in aqueous solution arising from photocorrosion under UV irradiation[7]. Both limitations are strongly correlated to the nature of the defect sites in ZnO NPs, which are thought to act as charge traps within the nanostructure. Though many techniques have attempted to shed light on the structure and energetics of these defects, their exact nature at RT remains under debate[8].

Several studies have been performed on ZnO using steady-state and time-resolved luminescence[9–17], and transient absorption optical spectroscopy[6,9,10]. Luminescence spectra of RT ZnO consist of visible and UV emission bands, which encompass several sub-bands that are observed at low temperatures[16]. The UV band lies close to the band gap absorption (3.3 eV for bulk ZnO[4]) and is assigned to the radiative recombination of free excitons in hundreds of ps (Supplementary Table 1). In contrast, the visible emission, usually in the green, is associated with deeply trapped charge carriers. Its position depends on the particle size, suggesting that one of the band edges must be involved in the radiative transition[11,18]. There is evidence that the green emission is related to the concentration of oxygen vacancies ($V_O$)[9–12,18–21], although several alternatives, such as Zn vacancies ($V_{Zn}$)[22,23], ZnO divacancies ($V_{ZnO}$)[8], and Zn interstitial defects[15], have also been proposed. These trapping sites may be expected to be predominantly located in the surface region of the nanoparticles as the green emission is quenched by the addition of hole scavengers ($I^-$ or $SCN^-$)[9]. An excess of $Zn^{2+}$ has no effect on the green emission, further suggesting its connection to an anion vacancy, such as an oxygen vacancy ($V_O$). EPR studies on ZnO nanopowders have shown that oxygen vacancies containing one electron ($V_O^+$) are the predominant paramagnetic defects[20,24–26], and the concentration of this defect is directly correlated with the green luminescence, leading to the conclusion that $V_O^+$ is involved in the emission process. Based on these studies, along with temperature-dependence experiments[11,27], it was concluded that the green emission stems from a shallowly trapped electron, forming a large polaron in the conduction band (CB), recombining with a $V_O^{2+}$ defect site that has been created through trapping of a photogenerated hole at a pre-existing $V_O^+$ site in the surface region. This mechanism clearly depends on the native $V_O^+$ concentration within the NP. Theory has calculated the energies and structures of the various defects[28–32], but has not managed to resolve the nature of the green emission because the large variety of defects leads to many acceptor/donor combinations, which can match the experimental energies.

The aforementioned optical studies are neither element-specific nor structure-sensitive. To identify the nature of the photo-induced trapping sites in RT ZnO NPs, time-resolved X-ray absorption spectroscopy (XAS)[33] has proved ideal with its sensitivity to the electronic structure of the absorber and to the local geometry of its surroundings. It was used to investigate electron injection from an adsorbed dye into iron oxide NPs in solution[34,35], as well as colloidal solutions of bare and dye-sensitized anatase and amorphous $TiO_2$ NPs after photoexcitation[36–38]. Because time-resolved XAS directly probes changes in the unoccupied density of states (DOS), which includes the CB of semiconductors, these studies, but also previous ultrafast optical domain measurements[39–42], were sensitive to the excitation and trapping of the electrons. However, the hole has escaped observation in TMOs, due to the fact that the valence band is dominated by the oxygen 2p orbitals, making it almost impossible to detect by X-ray or optical absorption. To address this, XAS can be complemented by X-ray emission spectroscopy (XES) to provide information about the time evolution of the occupied DOS, corresponding to the semiconductor valence band (VB)[43]. In the vicinity of an absorption edge this technique is called resonant X-ray emission spectroscopy (RXES). To date, the small cross-section of the RXES processes in the hard X-ray regime and the lack of high-brightness-pulsed X-ray sources have limited time-resolved RXES measurements to only a few examples in the literature[44–47].

Here we use time-resolved Zn K-edge XAS and RXES with 80 ps time resolution to probe the nature and lifetime of the photoinduced charge carriers in an aqueous colloidal solution of 32 nm ZnO nanoparticles. All the signals are measured simultaneously using a compact dispersive X-ray emission spectrometer[48], shown schematically in Fig. 1. Our results, supported by theoretical simulations, show that 80 ps after photo-excitation, the hole is trapped at an oxygen vacancy in the NP forming a small polaron. The XES signals corresponding to the electron, confirm that it remains, in contrast to the hole, spatially delocalised in the CB[12]. Simulations of the time-resolved X-ray spectra allow us to unambiguously identify the hole trap as a doubly positively charged oxygen vacancy ($V_O^{2+}$).

## Results

**Optical luminescence.** The measured optical luminescence spectrum of our sample excited at 350 nm exhibits the typical band gap luminescence at ~380 nm and green fluorescence at ~570 nm (Supplementary Fig. 1). The band gap fluorescence dominates the visible emission, which hints at a low number of defects[9–11,21].

**X-ray absorption spectroscopy.** Figure 2 shows the Zn K-edge ground state absorption spectrum recorded in total fluorescence yield (TFY) mode, along with the transient spectrum recorded 80 ps after excitation at 355 nm. Dramatic changes are observed over the entire range, including the pre-edge region. The steady state and transient XANES and EXAFS spectra are shown in Fig. 3. The transient XANES (Fig. 3d) shows positive features below the edge at 9.662 and 9.667 keV and a strong minimum at 9.671 keV, which is the position of the maximum absorption in the ground state spectrum. The region above the edge is characterised by changes giving rise to strong positive and negative features. As expected from the $d^{10}$ configuration of the Zn atoms and confirmed by Supplementary Fig. 2, which shows the simulated transient difference spectra for different shifts of the edge going from −5 to +5 eV, the changes at the edge cannot be attributed to a change of oxidation state (reduction or oxidation) of the Zn atoms. The kinetic trace of the transient feature at 9.670 keV is

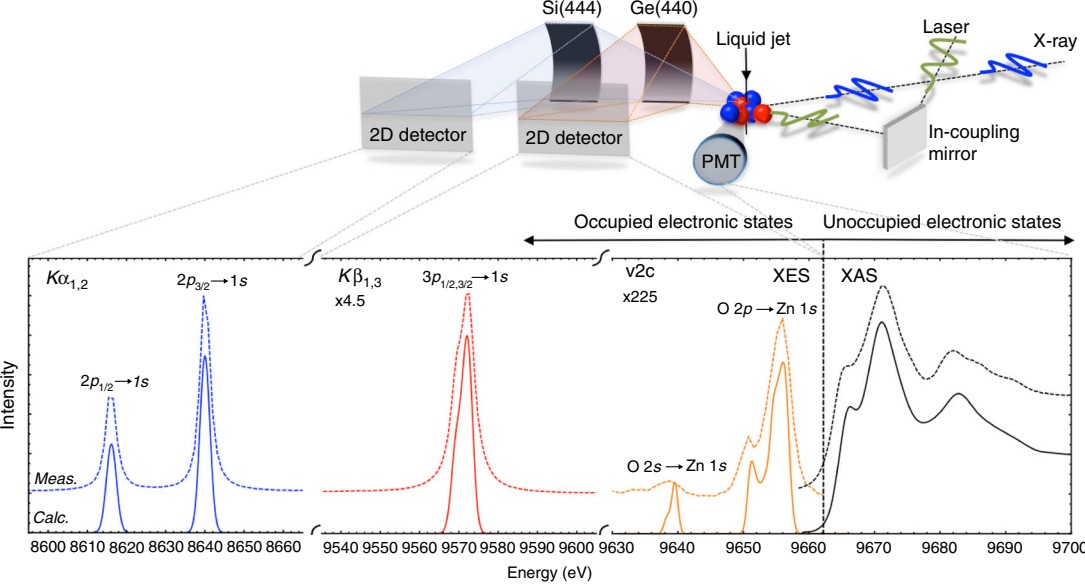

**Fig. 1** Experimental Setup. Upper panel shows: Schematic of the experimental setup used herein, with the two-crystal von Hamos XES spectrometer and scintillator/photomultiplier tube (PMT). The signals measured by the various components are shown adjacent to the detectors. Lower panel shows: The ground state Zn K-edge XAS, $K\alpha$, $K\beta$ and valence to core spectrum (dashed lines) and corresponding simulations (solid lines)

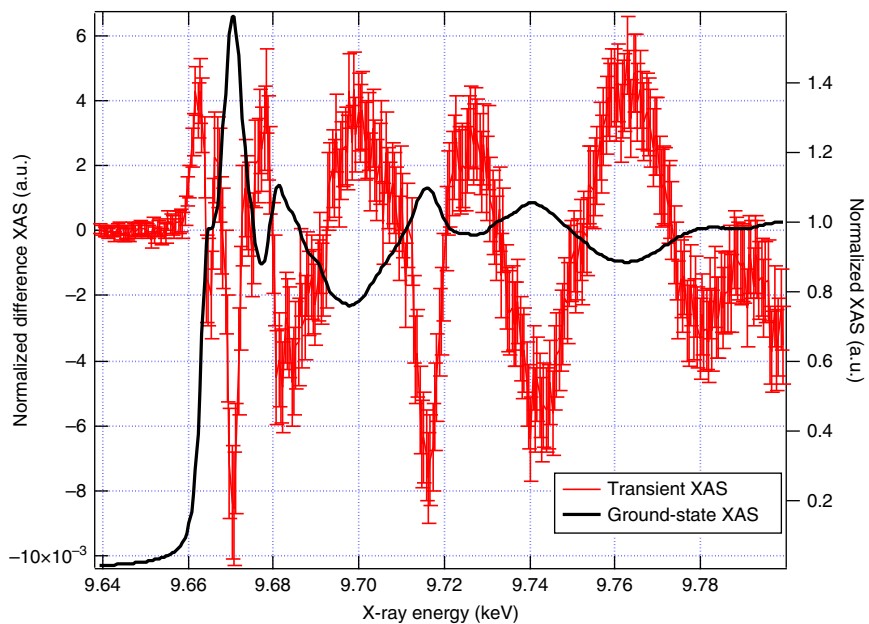

**Fig. 2** Ground-state and 80 ps transient X-ray absorption spectrum (XAS). Experimental ground-state XAS (black) and transient XAS difference of the excited minus the unexcited spectra (red) at 80 ps time delay after photo excitation with 355 nm. Error bars shown are the standard error of the experimental measurement

shown in Supplementary Fig. 3 along with its fit showing a pulse-width limited signal rise (~80 ps) followed by a biexponential recovery with decay time constants of $200 \pm 128$ ps and $1.2 \pm 0.3$ ns. These values fall within the range of literature values for the room-temperature photoluminescence (see Supplementary Table 1 for a summary). The short component (200 ps) reflects quite well the decay of the UV (band gap) luminescence[12,49,50], while the long one (1.2 ns) agrees well with the values reported for the green luminescence[9,10]. This strongly favours a decay stemming from the same initial state (large polarons or delocalised electrons in the CB) for both emission processes. The final state corresponds to the top of the VB in the case of the UV emission and to a deeply trapped hole state in the case of the green luminescence[17]. Both will have the same short component while the decay of the green luminescence contains an additional long component. The analogy of the kinetics of the X-ray signal with that of the green luminescence leads us to conclude that the former probes the traps that cause the green fluorescence. This is further supported by the analysis below. The conclusion that the electrons remain largely delocalised in the CB, as free carriers or large polarons is further supported by the XES results below.

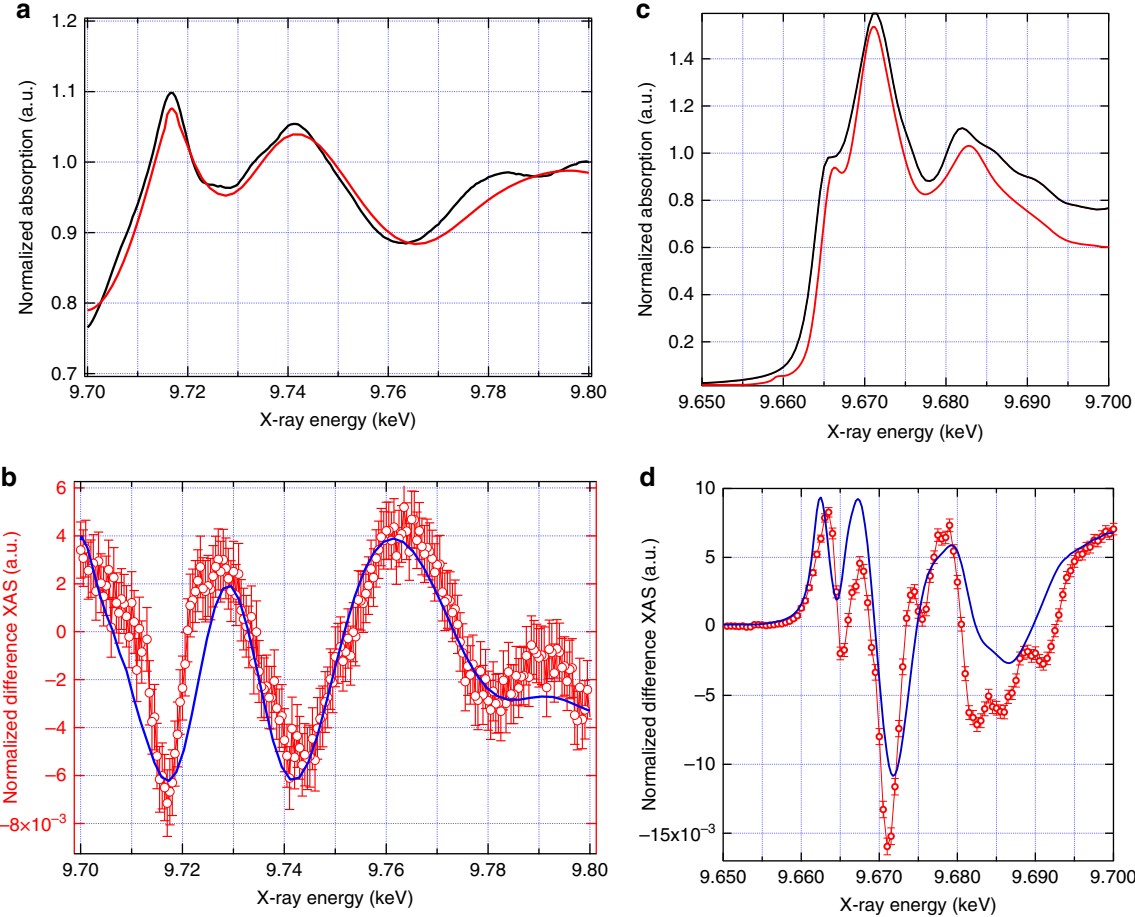

**Fig. 3** Ground-state and transient XAS spectra. **a** Experimental (black) and simulated (red) ground-state EXAFS spectrum. **b** The simulated difference EXAFS spectrum (blue) assuming an excited state structure corresponding to a $V_O^{2+}$ vacancy with 15% Zn–O bond distortion away from the vacancy and 13% excitation overlaid with the experimental transient EXAFS spectrum (red) measured at 80 ps after excitation. **c** Experimental (black) and simulated (red) ground-state XANES spectrum. **d** The simulated difference XANES spectrum (blue) assuming an excited state structure identical to that simulated in (**b**) and 13% excitation shown with the experimental transient XANES spectrum (red) measured at 80 ps after excitation. Error bars shown are the standard error of the experimental measurement

**Resonant X-ray emission spectroscopy**. Figure 4a, b show the ground state Zn $K\alpha$ and $K\beta$ RXES spectra. The $K\alpha$ emission shows the characteristic $K\alpha_1$ and $K\alpha_2$ lines arising from the Zn $2p_{3/2}$ (8.637 keV) and $2p_{1/2}$ (8.614 keV) levels, respectively. The $K\beta$ emission, stemming from the Zn $3p$ orbitals, shows a single broad emission band centred at 9.572 keV. Figure 4c, d show the transient difference $K\alpha$ and $K\beta$ RXES planes 80 ps after 355 nm photoexcitation. In both transient spectra, changes along the direction of the incident energy axis reflect those observed in the TFY transient XAS spectra, i.e., two positive features at 9.663 and 9.667 keV, followed by alternating negative and positive features.

Supplementary Fig. 4 shows the ground state and transient RXES spectra at incident X-ray energies of 9.663 and 9.8 keV. At 9.663 keV (Supplementary Fig. 4a, b), both the transient $K\alpha$ and $K\beta$ RXES spectra show a positive feature, indicating an increase in the emission intensity. In contrast, the transient $K\alpha$ and $K\beta$ RXES spectra at 9.8 keV display a broad negative feature, indicating a decrease in the emission signal. By comparing the transient $K\alpha$ and $K\beta$ RXES planes (Supplementary Fig. 4) with the transient TFY XAS signals shown in Figs. 2 and 3, it is clear that the changes in the transient RXES spectra (Fig. 4 and Supplementary Fig. 4) are dominated by the changes in the absorption cross sections at these incident X-ray energies. Here one must recall that the RXES cross-section is a convolution of the XAS and XES

transition matrix elements (see Supplementary Note 1 in the Supplementary Information for details). Crucially, this implies that charge carrier trapping does not result in strong changes of the Zn 3d density of states, which are expected to most strongly modulate the $K\alpha$ ($2p \rightarrow 1s$) and $K\beta$ ($3p \rightarrow 1s$) emission through the 2p–3d and 3p–3d exchange integrals, respectively[51], in line with the lack of an edge-shift in the transient XAS spectrum (Supplementary Fig. 2). Correcting for the change in the absorption cross-section (explained in Supplementary Note 3 of the Supplementary Information) reveals the purely XES contribution to the signal. This is shown in Fig. 5, where the derivative-like lineshapes reveal a slight shift of the emission signals, with the $K\alpha$ emission shifting to lower energies and the $K\beta$ emission shifting to higher energies. This effect is discussed in more detail below.

## Discussion

The ground state spectrum was already analysed and discussed in ref. [52]. It was found that zinc vacancies do not lead to measurable distortions of the local lattice structure, while the Zn K-edge XANES spectra show two weak pre-edge features, which were assigned to a small concentration of oxygen vacancies. Dileep et al. studied the effect of the $V_O$ concentrations using high-resolution electron energy loss spectroscopy (HREELS) at the

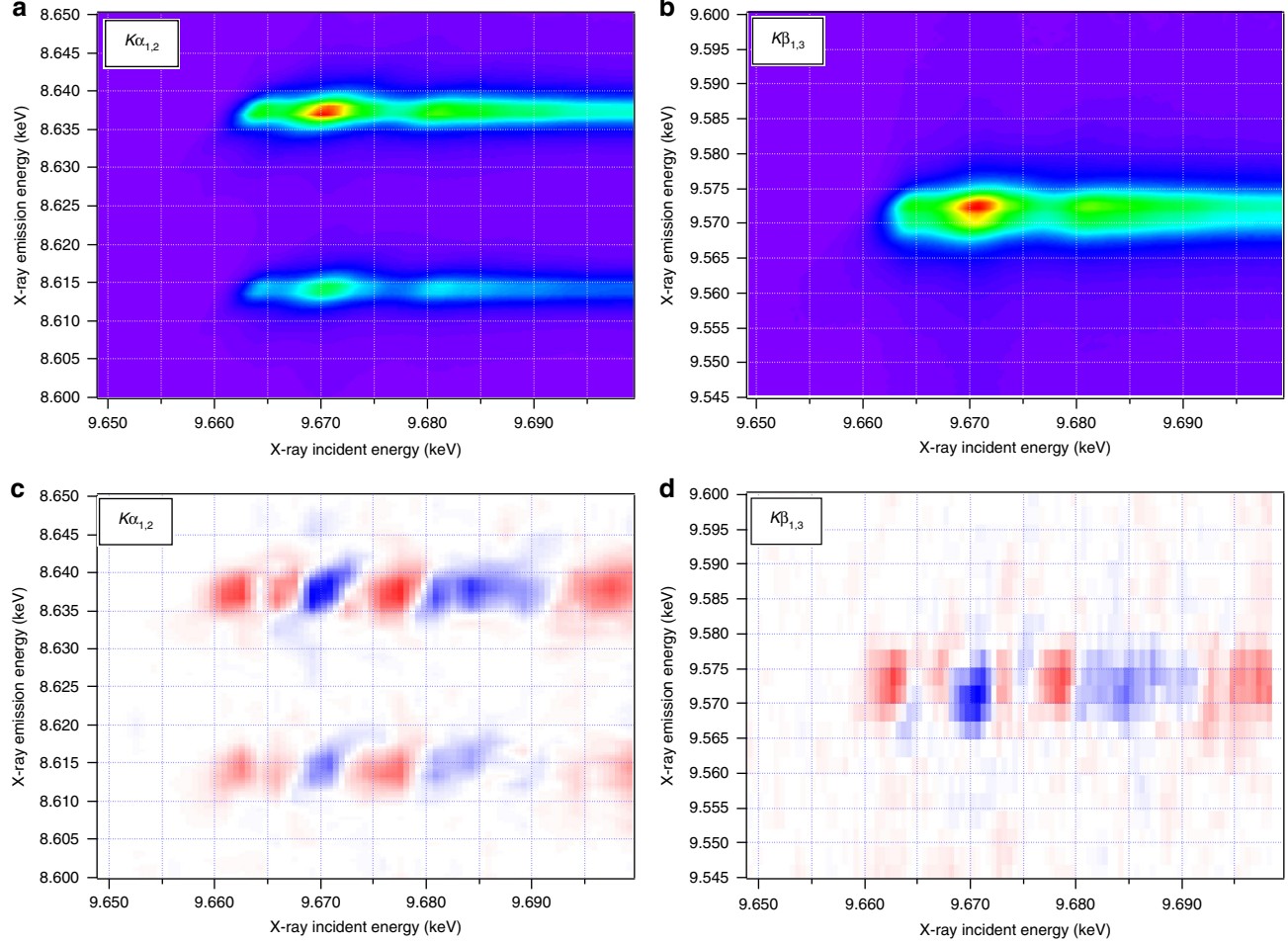

**Fig. 4** Ground state and transient $K\alpha$ and $K\beta$ RXES. The ground state Zn $K\alpha$ (**a**) and Zn $K\beta$ (**b**) RXES spectra of 32 nm ZnO nanoparticles in aqueous colloidal solution. The transient Zn $K\alpha$ (**c**) and Zn $K\beta$ (**d**) RXES spectra of 32 nm ZnO nanoparticles in colloidal solution measured 80 ps after photoexcitation at 355 nm with a fluence of 80 mJ/cm². In **c**, **d**, red indicates positive and blue indicates negative changes

Zn $L_3$-edge and the O K-edge[21]. The authors observed a larger pre-edge intensity in nanocrystals with a higher concentration of $V_O$, just as those observed at the Zn K-edge in ref. [52]. With the help of *ab initio* calculations, they assigned these new transitions to defect-related DOS occurring within the band-gap and exhibiting a mixed Zn $3d/4p$ and O $2p$ character. The transient spectrum (Fig. 2) shows clear changes at and above the absorption edge. However, as already mentioned, no discernible edge-shift (Supplementary Fig. 2) is seen indicating that the effective nuclear charge on the Zn atoms does not change significantly after photoexcitation, consistent with the $d^{10}$ configuration of the Zn atoms in ZnO. This implies that the electrons remain largely delocalised in the CB. Because the pre-edge feature at 9.663 keV of the transient is positive, it can only be due to a transition from the Zn 1 s orbital to a final Zn-$3d$/O-$2p$ hybridised orbital. Indeed, the creation of holes in the VB increases the density of unoccupied O $2p$ states with a consequence that the transition probability is increased in the pre-edge region of the Zn K-edge. The second positive feature at 9.667 keV corresponds to the $1s \rightarrow 4p$ transition in the ground state spectrum (Fig. 3)[52]. In several transition metal systems (molecules or NPs), this feature is sensitive to both hybridisation and to coordination numbers and distances, with a tendency to increase with increasing bond distances[53,54] or decreasing coordination number[55]. Finally, the decreased intensity right at the maximum reflects the effect of band filling due to the electrons that have been transferred to the

CB, thus reducing the transition probability above the Fermi level. The significant changes above the edge reflect significant structural modifications, as in the EXAFS region (Fig. 3). These structural changes cannot be approximated as a simple disorder-induced thermal effect, where the excited state spectrum can be approximated as a broadened ground-state spectrum, since agreement with the experiment is poor for this assumption (see Supplementary Fig. 10). Instead, these changes hint to a well-defined structure being generated after photoexcitation, which we discuss below.

The removal of an oxygen atom from a perfect ZnO crystal leaves four Zn dangling bonds. Janotti and van de Walle[28] calculated using Density Functional Theory (DFT), the local structure around oxygen vacancies in ZnO as a function of the charge at the vacancy. For $V_O^0$, the four Zn nearest neighbours are displaced inward by 12% of the equilibrium Zn–O bond length, whereas for $V_O^+$ and $V_O^{2+}$, the displacements are outwards by 2 and 23%, respectively. By contrast, zinc vacancies introduce partially occupied states close to the valence band minimum, arising from the broken bonds of the four oxygen nearest neighbours. However, all charge states of zinc vacancies exhibit a structural outward relaxation of ~10% with respect to the equilibrium Zn–O bond length[29]. We attempted to fit the transient EXAFS spectrum using the 2% $\left(V_O^+\right)$ and 23% $\left(V_O^{2+}\right)$ bond length elongations and the Zn vacancy ($V_{Zn}$) with and without the structural changes reported by Janotti and de Walle[28,29].

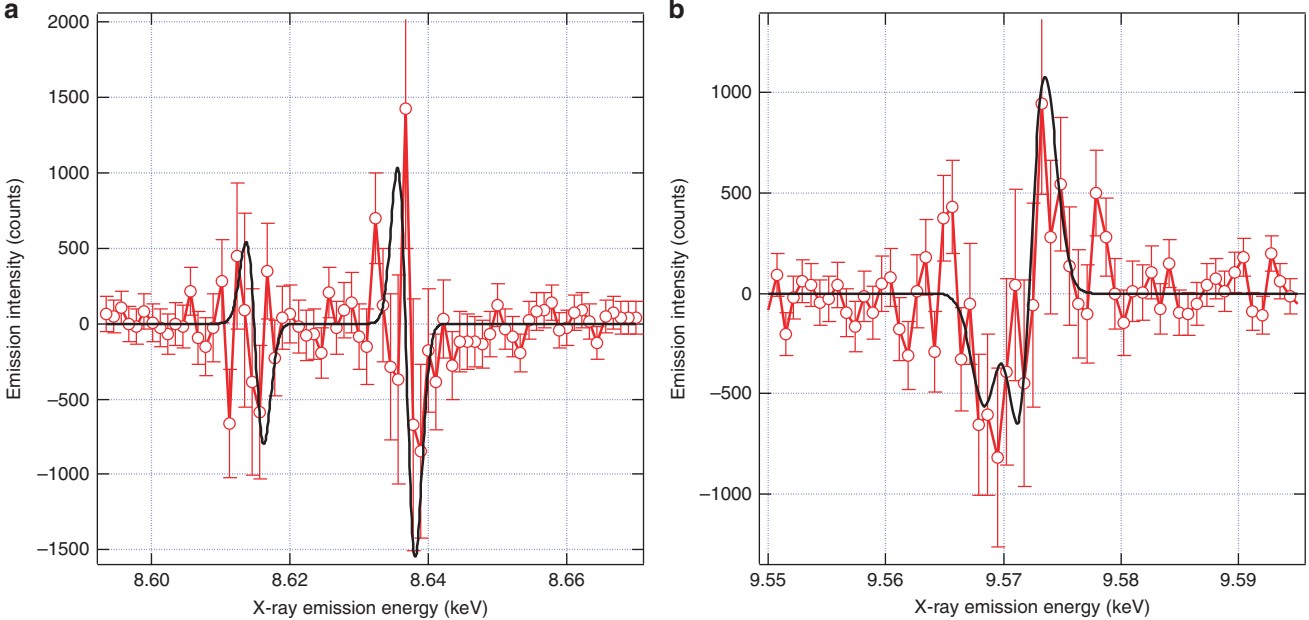

**Fig. 5** XES transients compared to simulation. Transient XES signals measured at an incident energy of 9.8 keV corrected for the change in total X-ray absorption cross-section (see Supplementary Note 3 for details) for the Zn $K\alpha$ (**a**) and $K\beta$ (**b**) emission signals (red data points) overlaid with the simulation of the transient XES signals using the excited state structure of the $V_O^{2+}$ vacancy (black curve). Error bars shown are the standard error of the experimental measurement

As shown in Supplementary Fig. 5, these give unsatisfactory agreement with the experimental transient.

We therefore followed an approach applied previously[56] by varying both the bond distance and the excitation yield to narrow down the best possible set of values that would reproduce the data. Throughout this work, the correlation between these parameters was found to be weak and therefore the refinement was performed by iteratively varying these two parameters independently. Further details on our analysis procedure can be found in Supplementary Note 2. On the basis of literature, the most probable defects are the singly or doubly charged oxygen vacancies[18–27]. Since the doubly charged oxygen vacancy structure provided the best agreement with the experimental transient, we used the $V_O^{2+}$ with a 23% lattice distortion to estimate the photolysis yield[28]. A photolysis yield of 7.5% gives the best agreement with the transient, as shown in Supplementary Fig. 6. Subsequently, by varying the Zn-Vacancy length, the best match in terms of phase and relative amplitude of the features was found for an outward distortion of 15% (Supplementary Fig. 7). This structure was then used to refine the excitation fraction for the new structure and obtain the best fit to the experimental transient. A photolysis yield of $13 \pm 3\%$ (Supplementary Fig. 8) was found, which is in good agreement with an estimate of 15% based on the measured optical absorption of the sample and is corroborated with a similar value retrieved from the fits of the XANES and XES transient signals.

The simulated EXAFS transients for the different defect structures, bond lengths and photolysis yields (Supplementary Figs. 5–8) show that the best agreement is achieved in the case of $V_O^{2+}$ with a 15% outward distortion and excitation fraction of $13 \pm 3\%$. The simulated transient and ground state EXAFS spectra are also reproduced in Fig. 3b along with the experimental data. This result implies that the hole capture occurs at a $V_O^+$ vacancy. What is remarkable to note from the present conclusions is that one single hole charge ($+e$) significantly displaces four Zn atoms by about 15% of the O-Zn distance in the regular lattice. This explains why the transient Zn K-edge XANES and

EXAFS signals are so strong, even though no charge localises on the Zn. The structure of the $V_O^{2+}$ photoinduced defect is compared to a perfect lattice site in Fig. 6.

Using the above parameters, we simulated the transient XANES spectrum for a $V_O^{2+}$ centre, which is shown in Fig. 3. This figure also reproduces the calculated static Zn K-edge spectrum from ref. [52] and the simulated and experimental transient XANES spectra are in good agreement. Supplementary Fig. 13 shows XANES simulations for other defect sites and the agreement is clearly poorer, which is consistent with the conclusions obtained from the EXAFS simulations. Thus, from Fig. 3 and Supplementary Figs. 5–8 and 13, we conclude that the long-lived trapped state and source of the green luminescence is the $V_O^{2+}$ state, in agreement with refs. [11,12] where the formation of $V_O^{2+}$ was hypothesised as a result of the trapping of a photogenerated hole at a pre-exisiting $V_O^+$ site.

The lack of an edge shift in our transient XAS spectra (Supplementary Fig. 2) is in agreement with the RXES results and imply that no significant change in the effective nuclear charge of the absorbing Zn atom is observed upon trapping of the hole. This is not unexpected because of the predominantly Zn 4s contribution to the unoccupied DOS of the CB[57], which results in more spatially delocalised states at the bottom of the CB. The lack of Zn 3d character in the CB is due to the $3d^{10}$ occupancy of the Zn atoms. In addition, the VB is of predominantly O 2p character with a small contribution from Zn 3d states (~9%[58]) due to hybridisation[59]. Therefore trapping of the hole at a $V_O$ does not introduce a significant shift of the Zn absorption edge. Our XES signals, when corrected for the change in X-ray absorption coefficient (see Supplementary Note 3 for further details), do show a small shift in emission energy. We simulated them using DFT (described in Supplementary Methods) for the $V_O^{2+}$ defect, and obtained remarkable agreement with the experimental measurement (see Fig. 5), further supporting our conclusion as to the nature of the hole trapping site in ZnO.

In contrast to previous observations in molecular systems[46,60], the transient $K\alpha$ and $K\beta$ spectra shift in opposite directions. The

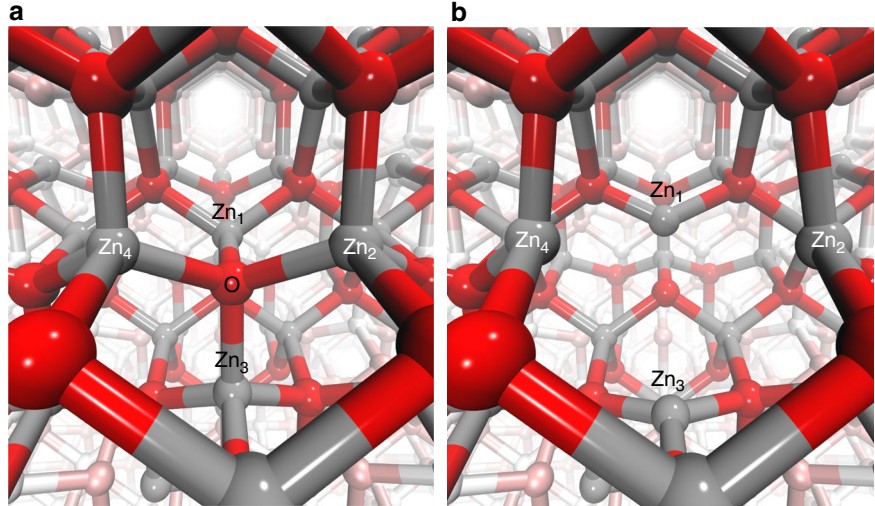

**Fig. 6** Structural changes. **a** Schematic of the ZnO structure without the oxygen vacancy, which is the dominant structure probed in the ground state, and (**b**) structural distortion around the $V_O^{2+}$ vacancy, which occurs upon the trapping of a hole at a $V_O^+$ vacancy and the corresponding outward displacement of the four nearest neighbour Zn atoms (Zn$_{1-4}$) by ~15%

origin of this observation lies in the competition between the two main factors influencing the emission energy of these spectra, namely the effective nuclear charge on the Zn atoms and the electron exchange interaction between the Zn $3d$ orbitals and the core levels, $2p$ or $3p$ for $K\alpha$ and $K\beta$, respectively. For the $K\alpha$, the transient spectrum shifts to the red, indicating a small loss of electronic charge density on the Zn atoms surrounding the defect site. This is confirmed by the DFT calculations, which accurately reproduce the observed experimental shifts, and indicate a 10% reduction in the electronic charge on the Zn atoms. This small change is consistent with the small experimental signal and the absence of any notable edge shift at the Zn K-edge XAS, which is dominated by much larger spectral features associated with the structural changes. It is also in line with the appearance of the pre-edge feature at 9.663 keV in the XANES spectrum. In contrast to $K\alpha$, the $K\beta$ XES spectrum shifts to higher energies. In this case, the emission line position is dominated by the $(3p, 3d)$ exchange interaction, which is sensitive to both the spin and covalency of the system[61,62]. Here, the structural change associated with the photoinduced hole trapping at the oxygen vacancy $(V_O^+ \rightarrow V_O^{2+})$ leads to a reduction of the covalency between the Zn $3d$ and O $2p$ orbitals. This shifts the spectrum to higher energies. Crucially, this is not observed for $K\alpha$ as the $(2p, 3d)$ exchange interaction is significantly weaker due to the smaller overlap of these wavefunctions[61,63]. This analysis, alongside the XANES and EXAFS data, supports the interpretation for hole trapping at an oxygen vacancy, creating a doubly charged vacancy. However, given the large error bars associated with the XES, it should not be used in isolation.

Finally, the hole trapping reported here is attributed to native oxygen vacancies which cannot be avoided in this material[52]. This conclusion is based on the theoretical modelling and on the analogy of the PL spectra and decay times with those of films or single crystals. However, from the present work we cannot distinguish between oxygen vacancies in the bulk and the surface shell. It is however obvious that the concentration of oxygen vacancies increases towards the surface shell, but as long as these are in the sub-surface region, their structure should be the same. At the surface itself, the structure is most likely different. In consequence, we primarily probe defects in the sub-surface to bulk regions. The fact that the valence band of TMOs is dominated by the O $2p$ orbitals suggests that the present results could

be extended to other TMOs. However, whether the reported hole trapping can serve as a model for other TMOs calls for further studies following the same approach as applied here.

In summary, using time-resolved Zn K-edge XAS and $K\alpha$ and $K\beta$ XES, we have probed the trapping of holes in aqueous colloidal dispersions of ZnO nanoparticles. Our results, supported by simulations, show a unique sensitivity to the hole via the structural changes it induces, demonstrating that it occurs at pre-existing $V_O^+$ defect sites, which become doubly charged $V_O^{2+}$ vacancies. This hole trapping results in an amplification of the signal due to the fact that four Zn atoms are significantly displaced to accommodate the hole polaron that is formed. The correspondence between the lifetimes of the X-ray signal and that of the green luminescence leads us to conclude that the final state of the latter is the $V_O^{2+}$ centre. Although ZnO exhibits a high electron mobility, making it attractive for devices demanding efficient electron transport, the correlation between transport and recombination processes associated with the $V_O^{2+}$ centre identified in this work, means that to tune device performance may require a suppression of the recombination dynamics, without having adverse effects on the transport properties. In this regard, ZnO nanorods have been shown to perform more efficiently as they reduce the concentration of oxygen vacancies[64].

The present study also shows the significant advantage of a compact dispersive X-ray emission spectrometer[48] for time-resolved measurements. Using such a spectrometer at hard X-ray-free electron lasers (XFELs), will permit RXES to be performed with femtosecond resolution, allowing the ultrafast charge carrier dynamics to be resolved. Indeed, the next step of the present study is to determine the timescale for hole trapping. Consequently, in order to fully address the kinetic discrepancy between the charge carrier lifetime and the solar energy or chemical fuel conversion, which is the limiting factor for many devices based upon TMO nanostructures, one must probe these dynamics on the femtosecond timescale, which has barely begun to be explored using structurally sensitive techniques[37,38].

## Methods
**Sample**. The zinc oxide nanoparticles were purchased from Sigma-Aldrich as colloidal dispersions in water. The measurements were performed on 170 mM solutions. The nanoparticles were determined to have a diameter of 32 nm. Further sample details can be found in ref.[52] and in the Supplementary Methods.

**X-ray spectroscopy measurements.** The experiment was performed at 7ID-D at the Advanced Photon Source, using a high-repetition rate laser and data acquisition setup[65]. The compact dual-von Hamos spectrometer was assembled at the beamline and used two Pilatus 100 K detectors to measure the dispersed $K\alpha$ and $K\beta$ X-ray emission spectra. The sample was excited with 10 ps pulses at 355 nm (3.5 eV) and a fluence of 80 mJ cm$^{-2}$. Further details are described in the Supplementary Methods.

**Calculations.** Calculations of the edge region of the spectrum were performed using the finite difference method as implemented within the FDMNES package[66], while simulations of the EXAFS region were performed using FEFF9[67]. The X-ray emission spectra were simulated using the one-electron approach[68] as implemented in the ORCA[69] quantum chemistry package. The computations are described in more detail in the Supplementary Methods.

**Data availability.** The data that support the conclusions of this work are available from the corresponding author upon reasonable request.

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

## Acknowledgements

G.D., A.M.M, and S.H.S. were supported by the U.S. Department of Energy, Office of Science, Basic Energy Sciences, Chemical Sciences, Geosciences, and Biosciences Division under Contract No. DE-AC02-06CH11357. This research used resources of the Advanced Photon Source, a U.S. Department of Energy (DOE) Office of Science User Facility operated for the DOE Office of Science by Argonne National Laboratory under Contract No. DE-AC02-06CH11357. M.C. acknowledges support by the Swiss NSF through the NCCR-MUST, by COST programs PERSPECTH2O (CM0702) and XLIC (CM1204), and the European Research Council Advanced Grant H2020 ERCEA 695197 DYNAMOX. A.B. and W.G. acknowledge the financial support from Hamburg Center for Ultrafast Imaging (University of Hamburg), Deutsche Forschungsgemeinschaft (via SFB925, TPA4), and the European XFEL GmbH. W.G. further acknowledges support from the National Science Centre Poland (NCN) under SONATA BIS 6 grant No. 2016/22/E/ST4/00543. C.J.M. acknowledges support by the Swiss NSF through the NCCR-MUST. T.J.P. acknowledges support from the Leverhulme Trust, grant number RPG-2016-103. J.S. acknowledges National Science Centre, Poland (NCN) for support under grant no. 2015/18/E/ST3/00444.

## Author contributions

J.S., S.H.S, W.G., G.D., R.A., M.C. and C.J.M. designed the experiment. J.S., F.G.S., A.B., W.G., G.D., A.M.M., S.H.S., J.R. and C.J.M. performed the experiment. T.J.P., J.S. and C.J.M. analysed the experimental data. T.J.P. performed the calculations. T.J.P., M.C. and C.J.M. wrote the manuscript with input from all authors.

## Additional information

**Competing interests:** The authors declare no competing financial interests.

