## [Peer Review File · Nature Communications]

Reviewer #1 (Remarks to the Author):

This is an interesting paper that tries to prove that hole trapping at oxygen vacancies is the dominant effect in the photoexcitation of ZnO nanoparticles. The paper utilizes X-ray absorption spectra, especially in the extended fine structure region, coupled with X-ray emission spectra, and with simulations, to argue that the mechanism for the spectral changes at 80 ps time is hole trapping at oxygen vacancies. The potential finding and its importance are meritorious for publication in Nature Communications. However, the proof of the outcome is not simple and the arguments are convoluted to arrive at the conclusion. Looking into detail in a number of figures, also in the supplement, a number of questions about the validation of the proof come up, which need to be addressed.

Figure 2 qualitatively looks like a differential change due to a bleach of the ground state spectrum as the main effect. The conclusion that there is no Zn oxidation state change appears sound. However, in order to see what changes are clearly different in this figure, the authors should be able to add back a fraction of the ground state spectrum to isolate the changes and make the fitting of just the changes more direct. Moreover, perhaps the excited state fraction can be determined independently with this procedure, whereas in the paper the excited state fraction seems to be another variable parameter in the fits.

Fig. S5 in the extended fine structure region seems to be one of the central arguments for the fact that the mechanism is hole trapping at the oxygen vacancy, with a 15% ZnO bond elongation. However, the black curve in this figure is not unique by any means, compared to the many other curves displayed, all of which consider no hole trapping at the oxygen vacancy. There are similar concerns about some of the other figures also, S2, S6, and Fig. 3 in the main manuscript. The main issue is that there is no real quantitative error analysis of the theoretical and fitting curves is performed to show how "sensitive" each of the fits is to separate variations in both the mechanisms (hole trapping, no hole trapping, oxygen vacancy, Zn vacancy) and the bond elongations and compressions.

The X-ray emission spectra represent one of the new aspects here, and the abstract notes that this is key to the arguments for hole transfer to the oxygen vacancy. A complex set of arguments is required to explain the directions of the spectral shifts, which depend on the changes in covalency of the orbitals, and the simulations in Fig. 5 are central to these arguments. However, no other simulations of the XES results are shown in the manuscript for the other possible mechanisms.

The lack of a singular overwhelmingly precise fit, together with possible missing checks against other mechanisms and more extensive parameter variations and error analysis, brings up the question whether there might be a combination of mechanisms at play. The paper seems to latch onto one potential explanation, but it does not seem to consider and then rule out each of the other possible mechanisms by extensive parameter variation and error analysis. Thus the conclusion, namely that hole trapping at oxygen vacancies is the main reason for the observations, does not seem to constitute a clear proof.

Reviewer #2 (Remarks to the Author):

In this manuscript, Milne et al. report state-of-art time-resolved X-ray absorption and emission spectroscopic measurements of aqueous ZnO nanoparticle colloidal solution, and on the basis of theoretical analyses, the authors proposed the photo-generated hole trapping as a positively charged oxygen vacancy, inducing significant structural displacements of neighboring Zn atoms. The X-ray spectroscopic RXES method (used in this work) combines the advantages of X-ray absorption (including EXAFS) and X-ray emission and allow the simultaneous observations of electronic structures (CB and VB) and local structural rearrangements. As authors noted, the

powerful methodology with XFEL at hard X-ray range could unambiguously unravel the ultrafast charge carrier dynamics of this important class of nanostructures. This paper is well written and the presentation is logical and scholarly. I generally like this type of work (a tour-de-force of high-level data handling methods applied to state-of-the-art measurements) and love to see such work appear in Nature Communications. I also think that this manuscript can draw the general interests (especially to the material science fields), although it contains highly complicated experimental scheme, analysis, and interpretation as well as its extensiveness. Therefore, I strongly recommend for publication of this manuscript in Nature Communications as long as the following few important points are addressed.

1. It is highly important for a complete understanding of the study to extend Fig. 3 to include the XANES experimental data and theory. This is also to show and discuss information on the electronic structure which is equally important for a consistent theory. It provides the reader with a better impression of how well (or not well) the theory works and how the reconstructed experimental spectrum looks like in order to follow the authors' interpretations such as the lack of a shift of the Zn K-edge. The current manuscript is not acceptable without it.

2. The excitation yield should be identical for EXAFS and XANES simulations for consistency. Despite the estimated error on the excitation yield, it will always be the same yield for both XANES and EXAFS data and hence the theory must work for the same value. The authors could show reconstructions for both, 11% and 13%. The current selective choice is inconsistent and speculative. It does not lend credibility to the interpretation.

3. With reference to point 1, the electronic structure should be discussed in more detail. It would be particularly important to explain the deviations between theory and experiment which only qualitatively describes the XANES region. Also, what is the Origin of the XANES difference signals that theory predicts and what are the conclusions for the changes in electronic structure, e.g. orbital hybridization, localization/delocalization of pDOS accessible by Zn K-edge spectroscopy, core-hole influence on the Zn 4p and 3d orbitals? The information is available from theory and should be presented in more detail.

Reviewer #3 (Remarks to the Author):

Penfold et al describe an x-ray spectroscopy study of hole relaxation dynamics in ZnO. They claim that their study serves as a model for hole trapping in other transition metal oxides (TMOs). The manuscript is generally well written, and the results are interesting. My main objection concerns the claim that ZnO is a model for TMOs. Unlike TMOs, the conduction band of ZnO is derived from Zn4s orbitals, hence its electron mobility is much larger than the for example TiO₂. This is reflected in the very different spectroscopies of the conduction band of the two materials.^{1,2} Generally the valence bands are derived from the O 2p orbitals, so there could be some useful lessons from ZnO for other TMO hole dynamics. Unfortunately the authors do not discuss comparisons with hole dynamics of other materials. Specifically, it is well accepted that in TiO₂ the holes are trapped by adsorbates^{3,4} under similar conditions as the experiment of Penfold et al, rather than the vacancy defects. This difference however is not discussed. The authors should provide some evidence that the solvent (H₂O) has no effect on the observed dynamics.

1 Deinert, J. C. et al. Ultrafast Exciton Formation at the ZnO(1010) Surface. Phys. Rev. Lett. 113, 057602 (2014).

2 Argondizzo, A. et al. Ultrafast Multiphoton Pump-Probe Photoemission Excitation Pathways in Rutile TiO₂. Phys. Rev. B 91, 155429 (2015).

3 Migani, A. et al. Level Alignment of a Prototypical Photocatalytic System: Methanol on TiO₂(110). J. Am. Chem. Soc. 135, 11429-11432, doi:10.1021/ja4036994 (2013).

4 Chu, W. et al. Ultrafast Dynamics of Photogenerated Holes at a CH₃OH/TiO₂ Rutile Interface. J.

Am. Chem. Soc. 138, 13740-13749, doi:10.1021/jacs.6b08725 (2016).

Reply to reviewers

Reviewer 1

This is an interesting paper that tries to prove that hole trapping at oxygen vacancies is the dominant affect in the photoexcitation of ZnO nanoparticles. The paper utilizes X-ray absorption spectra, especially in the extended fine structure region, coupled with X-ray emission spectra, and with simulations, to argue that the mechanism for the spectral changes at 80 ps time is hole trapping at oxygen vacancies. The potential finding and its importance are meritorious for publication in Nature Communications. However, the proof of the outcome is not simple and the arguments are convoluted to arrive at the conclusion. Looking into detail in a number of figures, also in the supplement, a number of questions about the validation of the proof come up, which need to be addressed.

Response: We thank the reviewer for their appreciation and thorough reading of our paper. We have performed several new theoretical calculations and experimental simulations of the excited state to address the reviewer's concerns.

- Figure 2 qualitatively looks like a differential change due to a bleach of the ground state spectrum as the main effect. The conclusion that there is no Zn oxidation state change appears sound. However, in order to see what changes are clearly different in this figure, the authors should be able to add back a fraction of the ground state spectrum to isolate the changes and make the fitting of just the changes more direct. Moreover, perhaps the excited state fraction can be determined independently with this procedure, whereas in the paper the excited state fraction seems to be another variable parameter in the fits.*

Response: The reviewer is correct, as asserted in the paper, that no Zn oxidation state change appears. This conclusion has been reiterated using a *shifted difference spectrum* for a range of different edge-shifts shown in **Figure R1**. This approach is commonly used to identify oxidation state changes. As observed in figure R1, none are able to reproduce the changes in the near edge region of the spectrum. This figure has been added to the supporting information.

Figure R1: Comparison between the 80 ps Zn K-edge experimental transient XANES spectrum and the shifted difference spectrum of ((ground state spectrum + shift eV) minus ground state spectrum). Although

for negative shifts, the simulated shifted difference spectrum captures the first two positive transient features, an edge shift leads to a derivative profile and therefore means that the simulation would also have to capture the negative feature to demonstrate that the spectral features derives from an edge-shift. This is not the case, and therefore an edge-shift can be excluded.

We have followed the reviewer's suggestion regarding the retrieval of the excited state spectrum. **Figure R2** below shows its derivation from the ground state spectrum and the transient signal for a range of excitation yields using:

$$A_{ES} = \frac{\Delta A}{f} + A_{GS}$$

Where A_{ES} is the excited state absorption, A_{GS} is the ground state absorption, ΔA is the transient and f is the excitation yield.

Figure R2: (a) Ground state spectrum (black) and excited state spectra extracted from the transient for various excitation yields. (b) A zoom of the spectral features occurring above the rising edge, with lower unrealistic photolysis yields removed.

The features occurring in the excited state spectrum remain strong, but slightly damped compared to the ground state. This is consistent with a reduced coordination of the Zn atoms but also the retention at a well-defined lattice structure around the trap sites. The referee is correct that upon first inspection the excited state signal appears to be a loss of ground state signal (i.e. the excited state spectrum appears closer to atomic Zn). Indeed, this accounts for a significant component of the transient signal, for which both positive and negative parts below, at and above the edge. However, if the excited state spectrum were purely a bleach of the ground state spectrum, the transient would be effectively simulated by the difference between a broadened ground state spectrum (to approximate the excited state) and the original ground state. This simulation is shown in **Figure R3**, for an excited state spectrum approximated using a Gaussian broadening with 3 different Full-Width-Half-Maximum (FWHM) values. Importantly, the transient features in the EXAFS region are not captured at all. Although the oscillations of the transient spectra in the XANES region are captured to some extent, there are important deviations, notably around 9.68-9.70 keV. In addition, the relative heights of the peaks are poorly reproduced. These differences encode the structural changes, which are analyzed in the paper.

Figure R3: (a) Ground State Zn K-edge spectrum of 32 nm ZnO nanoparticles in aqueous solution shown with Gaussian-broadened ground state spectra (Full-Width-Half-Maximum=5, 8 or 10 eV) to simulate possible bleaching and disorder in the excited state. (b) Transient Zn K-edge spectrum of 32 nm ZnO nanoparticles 80 ps after excitation at 355 nm (red trace) and the simulated spectrum assuming that the excited state spectrum is purely a broadened ground state spectrum, shown in (a). (c) The same simulated spectra shown in (b) in comparison to the XANES transient.

To clarify the discussion and provide a firmer base for the conclusions, **Figures R1 and R3** have been added to the supporting information.

2. *Fig. S5 in the extended fine structure region seems to be one of the central arguments for the fact that the mechanism is hole trapping at the oxygen vacancy, with a 15% ZnO bond elongation. However, the black curve in this figure is not unique by any means, compared to the many other curves displayed, all of which consider no hole trapping at the oxygen vacancy. There are similar concerns about some of the other figures also, S2, S6, and Fig. 3 in the main manuscript. The main issue is that there is no real quantitative error analysis of the theoretical and fitting curves is performed to show how "sensitive" each of the fits is to separate variations in both the mechanisms (hole trapping, no hole trapping, oxygen vacancy, Zn vacancy) and the bond elongations and compressions.*

Response: Throughout this work 2 variables were considered and varied in order to simulate the transient spectrum: bond lengths and photolysis yield. Throughout the work the correlation between these parameters was found to be weak. Indeed, a reduction in the coordination number reproduces all of features observed in the transient spectrum (See Figure S4, transient EXAFS spectrum) in more or less the correct positions.

Changes to the structural parameters (bond lengths) primarily modified the relative intensity of these features, with only small effects on the position of the peaks, while the effect of decreasing the photolysis yield is a constant damping across the whole transient signal, it therefore has no effect on the relative position and intensities of the transient features and can thus be obtained independently of the structural parameters.

The referee is correct that the analysis of the different trapping sites illustrated in Figure S4 (formerly S5) is central to the arguments of hole trapping. It is supported by the other measurements. This is the most appropriate approach as the physics required for an accurate description of the EXAFS spectrum is well established and leads to straightforward quantitative analysis of these spectra. This is in contrast to the XANES region of the spectrum where the complicated many-body physics often leads to only a qualitative agreement in many cases. The difference in the accuracy associated with each spectral region is compounded in time-resolved signals, where the transient is the subtraction of two spectra, with the possibility to contain 2 sets of errors.

Figure R4 shows each of the spectra shown in the original Figure S4 on separate plots. One can see that oxygen vacancy, with a 15% ZnO bond elongation (black trace) is the only curve that minimizes deviations with the experimental spectrum over the entire energy range. Important to note is that the deviations in the other cases are not simply a constant amplitude error, which could be account for by a different photolysis yield. The fact that the relative amplitudes and phases differs with respect to the experiment transient points to structural differences. From these V_{O}^{2+} is clearly the best fit, however to add rigour to this analysis, the root mean square deviation (RMSD) between the experiment and theory have been calculated and are given in brackets in the caption. These have been included in the supporting information for further information.

Figure R4: Experimental transient spectrum compared to the calculated transient EXAFS spectrum for the 5 different defect sites considered in this work: V_{O}^{2+} (black, RMSD = 0.0012), V_{O}^{+} (blue, RMSD = 0.0019), V_{O} (cyan, RMSD = 0.0035), V_{Zn} with structural change (pink, RMSD = 0.0027) and V_{Zn} without structural change (green, RMSD = 0.0023).

Finally, **Figure R5** shows the experimental transient XANES spectrum (red) compared to the calculated spectrum for the $V_{O^{2+}}$ (blue), V_{O^+} (green), V_{O^0} (orange), V_{Zn} (black). Here the transients for V_{O^+} , V_{O^0} and V_{Zn} are in poor agreement and consequently, can be ruled out. In support of the conclusions, this has been added to the supporting information as Figure S13 in the new submission.

Figure R5: Experimental transient XANES spectrum (red) compared to the calculated spectrum for $V_{O^{2+}}$ with 15% distortion (blue) V_{O^+} (green), V_{O^0} (orange), V_{Zn} (black).

3. *The X-ray emission spectra represent one of the new aspects here, and the abstract notes that this is key to the arguments for hole transfer to the oxygen vacancy. A complex set of arguments is required to explain the directions of the spectral shifts, which depend on the changes in covalency of the orbitals, and the simulations in Fig. 5 are central to these arguments. However, no other simulations of the XES results are shown in the manuscript for the other possible mechanisms.*

Response: The X-ray emission is an important supportive aspect to the interpretation. This, alongside the XANES and EXAFS data, support the interpretation for hole transfer to the oxygen vacancy. However, given the large error bars and the results discussed below, it certainly cannot be used in isolation. To address the referees' criticism, **Figure R6** shows the X-ray emission calculated for the zinc vacancy, without any structural distortion. Although the agreement for the transient $K\alpha$ spectrum is reasonable, especially given the large energy bars, the calculated transient $K\beta$ spectrum shifts to the red, in the same direction as the $K\alpha$ spectrum, which disagrees with the experimental results. This vacancy can therefore be discounted. This is because the zinc vacancy, being about 3.5 Å from the absorbing atom, is too far to influence the inherently short range (3p,3d) exchange interaction.

Figure R7 shows the transient $K\alpha$ and $K\beta$ spectra for the $V_{O^{2+}}$ (black) and V_{O^+} (blue) vacancy. The former is clearly in better agreement, although one should be careful of drawing too strong conclusions from this alone, due to the signal to noise ratio. However, this along with the XANES and EXAFS leads us the $V_{O^{2+}}$ vacancy conclusion.

Figure R6: Experimental K α (left) and K β transient spectra compared to the calculated spectra for a zinc vacancy.

Figure R7: Experimental K α (left) and K β transient spectra compared to the calculated spectra for two oxygen vacancies: V_O²⁺ (black) V_O (blue).

We have added these figures as Figures S11 and S12 to the supporting information.

4. *The lack of a singular overwhelmingly precise fit, together with possible missing checks against other mechanisms and more extensive parameter variations and error analysis, brings up the question whether there might be a combination of mechanisms at play. The paper seems to latch onto one potential explanation, but it does not seem to consider and then rule out each of the other possible mechanisms by extensive parameter variation and error analysis. Thus the conclusion, namely that hole trapping at oxygen vacancies is the main reason for the observations, does not seem to constitute a clear proof.*

Response: We hope with the above reply and the additional simulations we have included in the Supporting Information that we have convinced the referee that all our measured and simulated data is overwhelmingly consistent with our conclusions about the nature of the charge carrier trapping site.

Reviewer 2

In this manuscript, Milne et al. report state-of-art time-resolved X-ray absorption and emission spectroscopic measurements of aqueous ZnO nanoparticle colloidal solution, and on the basis of theoretical analyses, the authors proposed the photo-generated hole trapping as a positively charged oxygen vacancy, inducing significant structural displacements of neighboring Zn atoms. The X-ray spectroscopic RXES method (used in this work) combines the advantages of X-ray absorption (including EXAFS) and X-ray emission and allow the simultaneous observations of electronic structures (CB and VB) and local structural rearrangements. As authors noted, the powerful methodology with XFEL at hard X-ray range could unambiguously unravel the ultrafast charge carrier dynamics of this important class of nanostructures. This paper is well written and the presentation is logical and scholarly. I generally like this type of work (a tour-de-force of high-level data handling methods applied to state-of-the-art measurements) and love to see such work appear in Nature Communications. I also think that this manuscript can draw the general interests (especially to the material science fields), although it contains highly complicated experimental scheme, analysis, and interpretation as well as its extensiveness. Therefore, I strongly recommendation for publication of this manuscript in Nature Communications as long as the following few important points are addressed.

Response: We would like to thank the referee for their careful reading of the manuscript and their comments.

- 1. It is highly important for a complete understanding of the study to extend Fig. 3 to include the XANES experimental data and theory. This is also to show and discuss information on the electronic structure, which is equally important for a consistent theory. It provides the reader with a better impression of how well (or not well) the theory works and how the reconstructed experimental spectrum looks like in order to follow the authors' interpretations such as the lack of a shift of the Zn K-edge. The current manuscript is not acceptable without it.*

Response: We agree with the referee and have therefore added this in the revised version of the manuscript to Figure 3. Note in addition we have added Figure S2 to the supporting information to add support to our statement about the lack of shift of the Zn K-edge. Regarding the electronic structure – see point 3 below.

- 2. The excitation yield should be identical for EXAFS and XANES simulations for consistency. Despite the estimated error on the excitation yield, it will always be the same yield for both XANES and EXAFS data and hence the theory must work for the same value. The authors could show reconstructions for both, 11% and 13%. The current selective choice is inconsistent and speculative. It does not lend credibility to the interpretation.*

Response: In the original manuscript the excitation yield was $13\pm 3\%$, and for the EXAFS and XANES region and therefore, within the quoted error, the photolysis yields are consistent. However, we have made the photolysis yield 13% in both cases in the resubmitted manuscript, with the note that this has no discernible effect on the quality of the fit. In addition, the effect of the photolysis yield is a constant damping across the whole transient signal and therefore its effect can easily be evaluated by eye. It does not change the shape of the spectrum. Consequently, while we agree with the referee that the previous values were inconsistent, it cannot be described as speculative or lacking credibility because of the simple effect the excitation yield has on the transient spectrum.

3. *With reference to point 1, the electronic structure should be discussed in more detail. It would be particularly important to explain the deviations between theory and experiment, which only qualitatively describes the XANES region. Also, what is the origin of the XANES difference signals that theory predicts and what are the conclusions for the changes in electronic structure, e.g. orbital hybridization, localization/delocalization of pDOS accessible by Zn K-edge spectroscopy, core-hole influence on the Zn 4p and 3d orbitals? The information is available from theory and should be presented in more detail.*

Response: It is stressed that the majority of the structural features analyzed in the XANES region of the spectrum are above the rising edge and are therefore continuum resonances; consequently in this regime angular momentum (l) is no longer a good quantum number. This means analysis of the electronic structure in this manner has less meaning. Indeed, the Zn K-edge of ZnO displays no pre-edge in the ground state spectrum and only one in the transient, which is clearly due to the hole generated in the conduction band. All of the conclusions in this paper are drawn from the structural features.

Therefore the referee is correct in what the theory can bring regarding the electronic structure, however it has little meaning in the present context for ZnO and has therefore been left out, with greater emphasis placed upon the structural characteristics of the results. We anticipate that with other samples, measured under similar conditions, this type of analysis would indeed bring important further insight.

Reviewer 3

Penfold et al describe an x-ray spectroscopy study of hole relaxation dynamics in ZnO. They claim that their study serves as a model for hole trapping in other transition metal oxides (TMOs). The manuscript is generally well written, and the results are interesting. My main objection concerns the claim that ZnO is a model for TMOs. Unlike TMOs, the conduction band of ZnO is derived from Zn4s orbitals, hence its electron mobility is much larger than the for example TiO₂. This is reflected in the very different spectroscopies of the conduction band of the two materials. Generally the valence bands are derived from the O 2p orbitals, so there could be some useful lessons from ZnO for other TMO hole dynamics. Unfortunately the authors do not discuss comparisons with hole dynamics of other materials. Specifically, it is well accepted that in TiO₂ the holes are trapped by adsorbates under similar conditions as the experiment of Penfold et al, rather than the vacancy defects. This difference however is not discussed. The authors should provide some evidence that the solvent (H₂O) has no effect on the observed dynamics.

Response: In the original submission we do not claim that ZnO is a model for hole trapping in TMOs, merely that hole trapping has been hard to directly observe. The characteristics of X-ray spectroscopy and properties of ZnO provide a unique opportunity to directly observe this, but we do not claim or exclude that this mechanism is general. We agree with the referee that a discussion is needed in this context. Specifically regarding the role of surface adsorbates, we emphasise that the X-ray signal derives from every Zn atom in the sample and therefore, given the size of the particles, the contribution of the surface is likely to be limited. While we cannot exclude that these trapping effects occur at the surface, we are not directly sensitive to them. Indeed, in ref. J. Phys. Chem. C, **118**, 19422–19430 (2014) we showed that the Zn K-edge of these nanoparticles in water or ethanol gave identical signals. To address the concerns and directly address the role of surface effects, we have now added a text in the discussion that compares ZnO and TiO₂ (also given below). We hope that more insight will be delivered from studies with femtosecond resolution, which are ongoing.

'Finally, the hole trapping reported here is attributed to native oxygen vacancies which cannot be avoided in this material⁵¹. This conclusion is based on the theoretical modelling and on the analogy of the PL spectra and decay times with those of films or single crystals. However, from the present work we cannot distinguish between oxygen vacancies in the bulk and the surface shell. It is however obvious that the concentration of oxygen vacancies increases towards the surface, but as long as these are in the sub-surface region, their structure should be the same. At the surface itself, the structure is most likely different. In consequence, we primarily probe defects in the sub-surface to bulk regions. The fact that the valence band of TMOs is dominated by the O 2p orbital suggests that the present results could be extended to other TMOs. In TiO₂, an oxygen vacancy leaves 2 electrons behind, so the likelihood that a hole will be trapped at this type of vacancy is quite high. In EPR studies of TiO₂ powders under steady-state UV illumination, the EPR-detected holes produced by photoexcitation are O⁻ species, produced from lattice O²⁻ ions⁶³. Oxygen vacancies cannot be detected but the pattern of trapping at a doubly charged centre seems to be common to both ZnO and TiO₂. Furthermore, both materials exhibit a very similar green luminescence, which probably originates from the same oxygen vacancy. Therefore, we believe that the reported hole trapping can serve as a model for other TMOs. Further studies following the same RXES

approach as applied here will be necessary to establish if TMO charge carrier trapping trends can be established.'

Changes to paper draft:

1. Additional paragraph p. 14 bottom (text as above) in response to Reviewer #3
2. Modified *Figure 3* to include XANES results and simulations, as requested by Reviewer #2
3. Extended the statement on p. 7 about the lack to Zn XAS edge shift to read "As expected from the d^{10} configuration of the Zn atoms and confirmed by *Figure S2*, which shows the simulated transient difference spectra for different shifts of the edge going from -5 to +5 eV, the changes at the edge cannot be attributed to a change of oxidation state (reduction or oxidation) of the Zn atoms."
4. Modified affiliation of T.J.P. on p. 1 and SI p. 1
5. Added additional Acknowledgement sentence for J.S.
6. Added references supporting response to reviewers
7. Added mention of new SI figures into main text of the paper (see below for SI changes)

Changes to Supporting Information

Modified the SI figures in response to reviewers' comments and added references to the figures in the main text:

1. Removal of *Figure S2* since it is now reproduced in *Figure 3* in the main text
2. *Figure S2: Shifted Difference Spectrum* (new figure)
3. *Figure S5: Nature of the trapping site* now has separate plots for the various comparisons between experiment and theory and the caption includes the RMSD values for the presented curves. The only simulated curves presented are those using structures directly from Janotti et al. (references 1 and 2 in the SI).
4. *Figure S6: Excitation fraction dependence of the V_O^{2+} defect* compares various excitation fractions using the structure of the DFT-calculated V_O^{2+} defect, which had the best agreement in *Figure S5*
5. *Figure S7: Determining the Zn-Vacancy distortion* Comparison between various distortion distances for an excitation fraction of 7.5%
6. *Figure S8: Determining the Excited State Fraction* Checking the excitation fraction dependence of the signal obtained with a Zn-Vacancy distortion of 15% (from *Figure S7*)
7. *Figure S10: Estimating the Transient* Testing the hypothesis that the excited state XAS is merely a broadened version of the ground-state XAS
8. *Figure S11: Structural Sensitivity of the XES spectra* Comparison of experimental XES signals with simulations for V_O^+ and V_O^{2+} geometries
9. *Figure S12: Structural Sensitivity of XES spectra* Comparison of experimental XES signals with simulation for the V_{Zn} geometry
10. *Figure S13: Simulations of the Transient XANES Spectra* Comparison of experiments to XANES simulations of different defect geometries

Reviewer #1 (Remarks to the Author):

The authors have made good improvements to the manuscript, mainly through the supplemental figures. The frank discussion in the reply letter does not seem to translate to similar changes in the manuscript or the supplement, only the addition of many new figures. Thus readers may not realize the limitations of the claims and the authors have not nuanced their assertions as they do in the reply letter. The addition of some error analyses have been made, although a full systematic variation of fitting parameters has been avoided in the reply.

Reviewer #2 (Remarks to the Author):

In the current version of revised manuscript by Milne et al., the authors resolved all issues the reviewer raised. The reviewer would like to recommend this manuscript in the Nature Communications.

Reviewer #3 (Remarks to the Author):

The authors still make misleading comparisons between ZnO and TiO₂ in the revised manuscript. The fact that the two materials have similar optical emission spectra under some circumstances is not a valid basis for comparison. The emission from TiO₂ depends on the material and its environment (<http://pubs.acs.org/doi/abs/10.1021/jp8039934>). Furthermore, the trapped electrons in TiO₂ are believed to be interstitial Ti ions rather than O atom vacancy defects (<http://link.aps.org/doi/10.1103/PhysRevB.92.045204>), though in principle both could be present. It is also not correct that "Oxygen vacancies cannot be detected but the pattern of trapping at a doubly charged centre seems to be common to both ZnO and TiO₂. There is substantial evidence that the vacancies in TiO₂ are ionized (2+) with electrons loosely associated with them as polarons (<http://dx.doi.org/10.1002/pssr.201206464>) (<http://link.aps.org/doi/10.1103/PhysRevB.92.075308>). Thus, the doubly ionized vacancies would not be favorable trap states for holes. I find the comparison between ZnO and TiO₂ misleading and confusing, so cannot recommend publication in the present form.

Reply to reviewers

Reviewer #1:

The authors have made good improvements to the manuscript, mainly through the supplemental figures. The frank discussion in the reply letter does not seem to translate to similar changes in the manuscript or the supplement, only the addition of many new figures. Thus readers may not realize the limitations of the claims and the authors have not nuanced their assertions as they do in the reply letter. The addition of some error analyses have been made, although a full systematic variation of fitting parameters has been avoided in the reply.

Response: We thank the referee for recognizing the improvements made to the manuscript. Regarding the 'frank discussion' we can only identify our paragraph discussing the potential limitations of XANES analysis in contrast to the EXAFS analysis. This is well known amongst the X-ray community, but admittedly may not be the case amongst the broad readership expected of Nature Communications. We have therefore included this discussion in the theory section of the supporting information. The only other analysis which is nuanced with caution is the X-ray emission, due to the sizeable error bars. We felt this was clear that caution should be exercised because of the error bars, however to be completely clear on this issue we have added the following text in the penultimate paragraph of the analysis section in the main paper:

This analysis, alongside the XANES and EXAFS data, supports the interpretation for hole transfer to the oxygen vacancy. However, given the large error bars associated with the XES, it should not be used in isolation.

Finally regarding the systematic variation of the fitting parameters. We reiterate that initially we studied a large number of potential defect sites with transient XANES and EXAFS analysis and from this identified the most likely (Figure S5 and S13). Indeed, especially for the transient XANES analysis, the agreement between the experimental transient and simulations of defects other than the oxygen vacancy are poor. Once the oxygen vacancy had been established as the most likely defect site, two variables were considered and varied in order to simulate the transient spectrum. These were bond lengths and photolysis yield. These variables were systematically varied as shown in Figures S6, S7 and S8. The only systematic variation that we have not performed is an analysis in 2 dimensions, i.e. variation of both systematically at the same time. However this is only required when the correlation between the parameters is strong. Importantly, we reiterate, as discussed in the SI, that throughout the work the correlation between these parameters was found to be weak. Consequently such 2D analysis is not required. Indeed, the reduction in the coordination number reproduces all of features observed in the transient spectrum (See Figure S5, transient EXAFS spectrum) in more or less the correct positions. Changes to the structural parameters (bond lengths) primarily modified the relative intensity of these features, with only small effects on the position of the peaks, while the effect of decreasing the photolysis yield is a constant damping across the whole transient signal, it therefore has no effect on the relative position and intensities of the transient features and can therefore be obtained independently of the structural parameters.

In order to further clarify our analysis procedure we have added a Section to the SI (S5 Analysis of XAS results) where we describe in more detail the additional SI Figures and the information we included in our reply to the Reviewer.

Reviewer #2:

In the current version of revised manuscript by Milne et al., the authors resolved all issues the reviewer raised. The reviewer would like to recommend this manuscript in the Nature Communications.

Response: We would like to thank the reviewer for their efforts on this manuscript.

Reviewer #3:

The authors still make misleading comparisons between ZnO and TiO₂ in the revised manuscript. The fact that the two materials have similar optical emission spectra under some circumstances is not a valid basis for comparison. The emission from TiO₂ depends on the material and its environment (<http://pubs.acs.org/doi/abs/10.1021/jp8039934>). Furthermore, the trapped electrons in TiO₂ are believed to be interstitial Ti ions rather than O atom vacancy defects (<http://link.aps.org/doi/10.1103/PhysRevB.92.045204>), though in principle both could be present. It is also not correct that "Oxygen vacancies cannot be detected but the pattern of trapping at a doubly charged centre seems to be common to both ZnO and TiO₂. There is substantial evidence that the vacancies in TiO₂ are ionized (2+) with electrons loosely associated with them as polarons (<http://dx.doi.org/10.1002/pssr.201206464>, <http://link.aps.org/doi/10.1103/PhysRevB.92.075308>). Thus, the doubly ionized vacancies would not be favorable trap states for holes. I find the comparison between ZnO and TiO₂ misleading and confusing, so cannot recommend publication in the present form.

Response: The debate about trapped charges in transition metal oxides is very lively due to the many controversial and debated hypotheses. Our aim is not to contradict or endorse the referee's comments but we note that three out of the four papers she/he cites are theory papers, which do give hints but are not actual observations (while we cited experimental EPR studies in our paper), and two of these concern rutile TiO₂, while our comparison focussed on anatase TiO₂. Therefore the conclusion not to recommend publication because she/he disagrees with a part of the discussion is rather harsh. Thus, in order to avoid further controversies on these points that are not central to the paper, we have reformulated the paragraph in a more cautious way, avoiding definitive comparisons between TiO₂ and ZnO.

Reviewer #1 (Remarks to the Author):

The manuscript has addressed the concerns of this reviewer.